# Knockdown of microRNA390 Enhances Maize Brace Root Growth

**DOI:** 10.3390/ijms25126791

**Published:** 2024-06-20

**Authors:** Juan Meng, Weiya Li, Feiyan Qi, Tianxiao Yang, Na Li, Jiong Wan, Xiaoqi Li, Yajuan Jiang, Chenhui Wang, Meilian Huang, Yuanyuan Zhang, Yongqiang Chen, Sachin Teotia, Guiliang Tang, Zhanhui Zhang, Jihua Tang

**Affiliations:** 1National Key Laboratory of Wheat and Maize Crop Science/Collaborative Innovation Center of Henan Grain Crops/College of Agronomy, Henan Agricultural University, Zhengzhou 450046, China; juanjuanym@163.com (J.M.); liviya@126.com (W.L.); qi_fei_yan@163.com (F.Q.); jiangxiaoyu0604@163.com (N.L.); wan__jiong@163.com (J.W.); xiaoqili1998@163.com (X.L.); lojyj029@163.com (Y.J.); wchenhui2338@163.com (C.W.); huangmeilian11@163.com (M.H.); zhangyy216@163.com (Y.Z.); chenyongqiang@henau.edu.cn (Y.C.); 2Plant Molecular and Cellular Biology Program, University of Florida, Gainesville, FL 32611, USA; tianxiao.yang@ufl.edu; 3Department of Biotechnology, Sharda University, Greater Noida 201306, India; sachin.teotia1@sharda.ac.in; 4Department of Biological Sciences, Life Science and Technology Institute, Michigan Technological University, Houghton, MI 49931, USA; gtang1@mtu.edu; 5The Shennong Laboratory, Zhengzhou 450002, China

**Keywords:** maize, brace root growth, lodging resistance, miR390, auxin signaling, *ARF11*, *ARF26*

## Abstract

Brace root architecture is a critical determinant of maize’s stalk anchorage and nutrition uptake, influencing root lodging resistance, stress tolerance, and plant growth. To identify the key microRNAs (miRNAs) in control of maize brace root growth, we performed small RNA sequencing using brace root samples at emergence and growth stages. We focused on the genetic modulation of brace root development in maize through manipulation of miR390 and its downstream regulated *auxin response factors* (*ARFs*). In the present study, miR167, miR166, miR172, and miR390 were identified to be involved in maize brace root growth in inbred line B73. Utilizing short tandem target mimic (STTM) technology, we further developed maize lines with reduced miR390 expression and analyzed their root architecture compared to wild-type controls. Our findings show that *STTM390* maize lines exhibit enhanced brace root length and increased whorl numbers. Gene expression analyses revealed that the suppression of miR390 leads to upregulation of its downstream regulated ARF genes, specifically *ZmARF11* and *ZmARF26*, which may significantly alter root architecture. Additionally, loss-of-function mutants for *ZmARF11* and *ZmARF26* were characterized to further confirm the role of these genes in brace root growth. These results demonstrate that miR390, *ZmARF11*, and *ZmARF26* play crucial roles in regulating maize brace root growth; the involved complicated molecular mechanisms need to be further explored. This study provides a genetic basis for breeding maize varieties with improved lodging resistance and adaptability to diverse agricultural environments.

## 1. Introduction

Maize stands as one of the foremost vital crops, serving as a staple for human food and animal feed [1]. Over recent decades, significant enhancements in grain yield have been achieved through advanced breeding techniques and improved management practices, notably increasing plant density [2,3]. However, these improvements have also led to elevated risks of lodging, adversely impacting grain yield, quality, and escalating harvest costs [4]. Consequently, the development of lodging-resistant maize cultivars has emerged as a crucial objective among breeders. An optimal root system architecture is essential for augmenting resistance to lodging [5]. Lodging in maize is primarily categorized into stalk and root lodging. Stalk lodging typically manifests at late growth stages with internodes beneath the ear buckle, while root lodging can occur at any growth stage if the root system fails to anchor the plant upright [5,6,7,8,9]. The maize yield loss due to root lodging can be up to 30% [10]. Studies increasingly show that resistance to root lodging substantially depends on the root system’s size and extent [5,11]. The maize root architecture comprises embryonic primary, seminal, lateral, and nodal roots [8,9]. Among these, nodal roots, including subterranean crown and aerial brace roots, play pivotal roles in anchorage and water uptake, predominantly expressing diverse water and nutrient transporters [8,12]. The architecture of brace roots is intimately linked with their capacity to resist lodging [5,9,11,13]. Of the developmental phases of brace roots—induction, initiation, emergence, and growth—emergence and growth phases are critical for establishing the root’s angle and length, which are crucial for plant anchorage [8,12,14]. Thus, deciphering the molecular mechanisms underlying brace root emergence and growth stages is imperative for a deeper understanding of maize root architecture.

The characteristics of maize brace roots, such as whorls, the number of roots per whorl, diameter, and spread width, significantly correlate with lodging resistance [5,11,13,15]. Particularly, the spread width is closely associated with root length and root angle, shaped during brace root growth [5,8]. Despite extensive research, mutants like *rootless concerning crown and seminal roots1* (*rtcs1*) [16], *rtcs-like1* (*rtcl1*) [16], *lateral rootless1* (*lrt1*) [17], *auxin regulated gene involved in organ size8* (*argos8*) [18], *rootless1* (*rt1*) [19], and *big embryo1* (*bige1*) [20], have revealed their influence on brace root whorls, number, and emergence. Mutant genes like *rtcs-like1* (*rtcl1*) [21], *ZmYUC2* [11], *ZmYUC4* [11], *ZmRSA3.1* [15], *ZmRSA3.2* [15], and *ZmCIPK15* [22] were identified to associate with maize brace root growth and architecture. Yet, the comprehensive molecular mechanisms driving brace root growth remain largely elusive.

MicroRNAs (miRNAs), comprising 20–24 nucleotides, are endogenous small RNAs that play critical roles in gene silencing through transcript cleavage or translational repression [23]. These molecules are fundamental in regulating plant growth, reproduction, and biotic and abiotic stress responses. Recent studies have highlighted the influence of specific miRNAs, such as miR160, miR165/166, miR396, miR167, miR390, miR857, and miR169, in shaping plant root systems [24,25,26,27,28]. For instance, miR390 is particularly notable, regulating root growth and structure by triggering the production of *TAS3*-transacting siRNAs (tasiRNAs), which target *auxin response factors* (*ARFs*), influencing various developmental processes [24,26,29,30,31]. However, the specific roles of miR390 in maize brace root development remain unclear. In maize, the miR390 family has two members, miR390a and miR390b, which cleave *TAS3* precursors and generate tasiRNAs [26]. *TAS3*-tasiRNAs further target several *ARFs*, including *ZmARF11*, *ZmARF12*, *ZmARF23*, *ZmARF24*, and *ZmARF26* [32]. In the present study, we perform deep sequencing of small RNAs to identify miRNAs involved in brace root growth. By constructing miR390 knockdown vectors through short tandem target mimic (STTM) technology, we generated transgenic mutants to validate the functional roles of the identified miRNAs. Our findings suggest that miR390 significantly influences brace root development via the *TAS3*-*ZmARF11*/*ZmARF26* pathway, underscoring its potential as a target for genetic enhancements aimed at improving root system architecture and developing lodging-resistant maize varieties.

## 2. Results

### 2.1. Identification of Key miRNAs in Brace Root Growth

To identify the key miRNAs involved in regulating maize brace root growth, brace root samples from the inbred line B73 were collected at emergence and growth stages and subjected to small RNA deep sequencing and transcriptome analysis. This comprehensive approach led to the identification of 22 differentially expressed miRNAs, split into 12 up-regulated and 10 down-regulated miRNAs. Notably, seven known miRNAs, including zma-miR167b-3p, zma-miR172d-5p, zma-miR390b-5p, zma-miR390a-5p, zma-miR166m-3p, zma-miR172b-5p, and zma-miR166l-3p, were found to be up-regulated (Figure 1, Table 1). All ten down-regulated miRNAs were novel, suggesting they might play previously unrecognized roles in maize brace root development.

Transcriptome sequencing revealed that 3410 genes were differentially expressed between the emergence and growth phase of B73, with 1195 genes being up-regulated and 2215 genes down-regulated (Figure 1, Table 2). Among the up-regulated differentially expressed genes (DEGs), notable ones include *Zm00001d001960* (*flavanone 3-hydroxylase1*, *FHT1*, involved in response to light stimulus), *Zm00001d014914* (*anthocyaninless2*, *A2*, involving in anthocyanin biosynthesis), and *Zm00001d044122* (*anthocyaninless1*, *A1*, involved in asteroid biosynthetic process). Others, like *Zm00001d007403* (*white pollen1*, *WHP1*, involved in flavonoid biosynthetic process, auxin transport, and root gravitropism), *Zm00001d021991* (*udp-galactose transporter 2-like*, involved in nucleotide-sugar transport), *Zm00001d010630* (*skewed root growth similar4*, *SKUS4*, involved in lateral root growth), *Zm00001d039644* (*cytokinin-o-glucosyltransferase 3*, involved in cytokinin synthesis), and *Zm00001d009057* (involved in cell wall organization) also showed significant up-regulation expression, indicating their involvement in diverse biochemical pathways critical for root development.

Of the down-regulated DEGs, key phytohormone signaling- and cell wall organization-related genes showed marked down-regulation between emergence and growth phases of brace root development. These include *Zm00001d011156* (*polygalacturonase45*, *PGL45*), *Zm00001d044031*, *Zm00001d040379*, *Zm00001d049336*, *Zm00001d024178* (*xyloglucan glycosyltransferase2*, *XGT2*), and *Zm00001d050201*, which are essential for cell wall biogenesis and organization; *Zm00001d054055* (*walls are thin 1-like*, *WAT1*, involved in unidimensional cell growth and auxin efflux), *Zm00001d043244* (involved in auxin response), *Zm00001d031594* (involved in auxin efflux), *Zm00001d052112* (*GRF-transcription factor 10*, *GRFTF10*, involved in gibberellin response), *Zm00001d018260* (*GRF-transcription factor 9*, *GRFTF9*, involved in gibberellin response), and *Zm00001d020418* (*cytokinin hydroxylase-like*, involved in cytokinin synthesis), among others, highlighting the critical roles of phytohormone signaling in brace root growth and development.

In addition, transcriptome data revealed that the target genes of miR167, *Zm00001d001879* (*auxin response factors transcription factor 3*, *ZmARF3*), *Zm00001d041497* (*ZmARF9*), *Zm00001d053819* (*ZmARF16*), *Zm00001d014377* (*ZmARF18*), *Zm00001d036593* (*ZmARF22*), *Zm00001d026590* (*ZmARF30*), and *Zm00001d031064* (*ZmARF34*) [33] showed sightly up-regulated expression. Known target genes of miR172 [34], including *Zm00001d019230* (*sister of indeterminate spikelet1*, *sid1*) and *Zm00001d046621* (*gloosy15*, *gl15*), showed significant down-regulated expression; *Zm00001d034629* (*tasselseed6*, *ts6*) and *Zm00001d035512* (*AP2-EREBP-transcription factor 81*, *ereb81*) displayed insignificant different expression. Moreover, downstream regulated *ARFs* of miR390 [32], including *Zm00001d043431* (*ZmARF1*1), *Zm00001d043922* (*ZmARF12*), *Zm00001d038698* (*ZmARF23*), *Zm00001d039006* (*ZmARF24*), and *Zm00001d012731* (*ZmARF26*), showed slightly up-regulated expression, suggesting a complex regulatory network influenced by miR390 in maize brace root development. Lastly, miR166 target genes, such as *Zm00001d048527* (*rld1*), *Zm00001d027317* (*rld2*), *Zm00001d031061* (*Homeobox-transcription factor 119*, *ZmHB119*) and *Zm00001d041489* (*ZmHB25*) displayed varied expression patterns during brace root growth, further underscoring the sophisticated modulation of brace root architecture by miRNAs.

### 2.2. Enhancement of Brace Root Growth in Maize via STTM-Mediated Knockdown of miR390

Previous research has suggested that established short tandem target mimic (STTM) technology is highly effective for investigating miRNA function [35,36]. To explore the functions of miRNAs identified as crucial for brace root growth, miR390 was specifically targeted for knockdown using an STTM vector constructed with a 48nt linker sequence and two un-cleavable miR390 binding sites. Each binding site was designed to include a three-base insertion (CTA) between the 10th and 11th bases, ensuring resistance to miR390 cleavage (Figure 2A). The obtained *STTM390* transgenic mutant plants exhibited notable phenotypic changes, including reduced plant height, narrower leaf angles above the uppermost ear, and broader leaf angles below the uppermost ear compared to wild-type plants, which typically show narrow leaf angles below the uppermost ear (Figure 2B). The relative expression levels of miR390 in these *STTM390* mutants ranged from 21.2% to 55.1% compared to the control line C01, demonstrating effective knockdown (Figure 2C).

Analysis of *TAS3* precursor expressions revealed a significant decrease in *TAS3c* expression in *STTM390* plants, whereas *TAS3g* expression was upregulated (Figure 2D). This differential expression pattern suggests a complex regulation by miR390 on different *TAS3* precursors. The downstream target genes of *TAS3*-tasiRNAs, including *ZmARF11* and *ZmARF26*, displayed up-regulated expression, aligning with the observed phenotypic alterations. Remarkably, the *STTM390* plants displayed a narrow brace root angle and increased brace root whorls (Figure 2E–G). When comparing with the length of the first layer brace root, those of *STTM390* mutants were more than twice as long as those in C01. These findings strongly suggest that miR390 plays a critical role in regulating both brace root growth and the gravimetric response in maize, potentially offering new avenues for enhancing root architecture through targeted miRNA manipulation.

### 2.3. Contrasting Roles of ZmARF11 and ZmARF26 in Enhancing Brace Root Growth and Impeding Soil Penetration in ZmARF EMS Mutants

In the *STTM390* transgenic maize, the expression levels of miR390 downstream genes, *ZmARF11* and *ZmARF26*, were significantly up-regulated in brace roots, suggesting their active roles in brace root development. To further investigate these roles, EMS mutants *arf11* (*EMS4-10a34d*) and *arf26* (*EMS4-1c4e8c*) were analyzed under field conditions. Compared to the wild type (B73), both *arf11* and *arf26* mutants exhibited a slight increase in plant height, with no notable difference in leaf shape and stalk thickness (Figure 3). Phenotypically, the *arf11* mutant plants demonstrated an increase in the number and length of brace roots; however, the penetration of these brace roots into the soil was reduced. Conversely, the *arf26* mutant plants had increased brace root length and diameter but a decreased number of brace roots that enter the soil. These observations suggest that *ZmARF11* and *ZmARF26* play distinct roles in modifying brace root architecture. While these genes clearly contribute to physical changes in root traits, the precise molecular mechanisms by which they influence brace root growth require further elucidation. This continued investigation is essential to fully understand the regulatory pathways involved in brace root development mediated by miR390 and its downstream regulated genes.

### 2.4. Alterations of Brace Root Microstructure and Phytohormone Levels in STTM390

The vascular bundle, a crucial transport system of vascular plants, mainly comprises phloem and xylem, serving essential roles in nutrition transport and structural support for the aerial parts of the plant [37,38]. To investigate the microstructural changes in brace roots of maize *STTM390* mutants, histological analyses were conducted on the cross sections from the elongation zone of brace roots in both C01 and *STTM390* plants (Figure 4). The analysis revealed a significant reduction in the number of xylem vessels in maize *STTM390* plants compared to C01, with the remaining vessels exhibiting uneven distribution. Additionally, there was a noticeable decrease in the thickness of the brace root phloem in *STTM390* plants, suggesting that the STTM-mediated knockdown of miR390 led to substantial alterations in the microstructure of maize brace roots.

Phytohormones such as auxin are known to play pivotal roles in brace root development, and miR390 has been implicated in auxin signaling pathways. Accordingly, the levels of indole-3-acetic acid (IAA) and abscisic acid (ABA) were measured in both *STTM390* and C01 plants (Figure 4). The results indicated a significant increase in ABA content and a decrease in IAA levels in *STTM390* plants. These changes in phytohormone levels are indicative of the regulatory influence of miR390 on hormone synthesis or signaling pathways, further affecting root development and plant response to environmental stimuli.

### 2.5. Unexpected Regulatory Interactions: Downregulation of Both miR166 and Its Targets following miR390 Knockdown in Maize

Previously studies established an interaction between miR166 and miR390 through the *TAS3*-tasiRNA pathway in maize [39,40]. To further explore the impact of the miR390 knockdown on miR166 and its target genes, we conducted RT-qPCR analysis on several key genes regulated by miR166. These genes include *Zm00001d031061* (*Homeobox-transcription factor 119*, *ZmHB119*), *Zm00001d041489* (*ZmHB25*), *Zm00001d048527* (*Rolled leaf1*, *Rld1*), and *Zm00001d027317* (*Rld2*) (Figure 5). The RT-qPCR results indicated a down-regulation in the expression of miR166 as well as all its analyzed target genes in *STTM390* plants compared to controls. This down-regulation suggests that the knockdown of miR390 may indirectly influence the expression patterns of miR166 and its associated regulatory network, impacting developmental and morphological processes governed by these genes. These findings contribute to our understanding of the complex regulatory relationships between miRNAs in maize and underscore the broader implications of miR390 manipulation on the maize transcriptome, particularly in how it influences other significant miRNAs and their downstream effects.

### 2.6. Transcriptomic Profiling in miR390 Knockdown Maize Reveals Dysregulation in Senescence, Cell Wall Biogenesis, and Phytohormone Signaling Pathways

To elucidate the regulatory pathway influence by miR390 in maize brace root growth, transcriptome analysis was conducted on brace root samples from both *STTM390* and C01 (Table 3). The transcriptome analysis identified a total of 294 differentially expressed genes (DEGs), with 244 genes showing up-regulation and 50 genes showing down-regulation in the *STTM390* mutants compared to controls. Validation through RT-qPCR confirmed the expression trends observed in the transcriptome data for selected genes (Figure 6, Table 3).

Among the up-regulated DEGs, several were identified as key players in various bio-logical processes, such as wounding response, senescence and phytohormone response, lignin catabolism and microtubule function, and cell death. Wounding response: genes such as *Zm00001d053934*, *Zm00001d053933,* and *Zm00001d015077*, which are implicated in the plant’s response to physical damage, showed increased expression in *STTM390* plants. Senescence and phytohormone response: genes such as *Zm00001d004355*, *Zm00001d037303*, *Zm00001d045711*, and *Zm00001d045708* also showed up-regulated expression in *STTM390* plants. Lignin catabolism and microtubule function: A gene involved in lignin breakdown (*Zm00001d049085*) and two genes related to microtubule-based movement (*Zm00001d011129* and *Zm00001d021253*) exhibited increased expression levels. Cell death: three genes linked to cell death processes (*Zm00001d047256*, *Zm00001d004355,* and *Zm00001d027486*) were identified to display up-regulated expression in *STTM390*.

Furthermore, genes involved in jasmonic acid biosynthetic processes (*Zm00001d052096*), gibberellin biosynthetic processes (*Zm00001d047830*), and plant-type cell wall organization (*Zm00001d034551*) also displayed up-regulation in *STTM390* plants. Conversely, two cell wall biogenesis-related genes (*Zm00001d039231* and *Zm00001d027525*), an ethylene biosynthesis-related gene (*Zm00001d049823*), and an auxin-responsive protein encoding gene (*Zm00001d021457*) were among those that showed reduced expression levels in *STTM390.* Significant changes were also observed in auxin signaling-related genes between the *STTM390* and C01 samples, highlighting genes like *Zm0001d030310* (*lax1*), *Zm0001d016277* (*IAA7*), *Zm0001d045203* (*IAA29*), and *Zm0001d018024* (*PIN2*), along with *ZmARF11* and *ZmARF26*, which showed notable expression changes.

Gene Ontology (GO) analysis indicated that most DEGs were associated with molecular functions, with fewer genes enriched in cellular components or biological processes (Figure 7). The down-regulated DEGs were mainly linked to purine ribonucleoside tri-phosphate binding, metabolic processes involving purine-containing compounds, and various aspects of cell wall macromolecule metabolism and biosynthesis. Up-regulated DEGs were predominantly enriched in activities involving catalysis, hydrolase and peptidase activity, proteolysis, and responses to ethylene, highlighting their roles in the cell periphery, plasma membrane, and extracellular regions. The Kyoto Encyclopedia of Genes and Genomes (KEGG) analysis further revealed that down-regulated DEGs were enriched in metabolic pathways related to secondary metabolites, purine metabolism, and plant hormone signal transduction, while up-regulated DEGs were more frequently associated with pathways involved in amino acid biosynthesis and pyrimidine metabolism (Figure 8). These findings provide valuable insights into the complex genetic and biochemical landscape altered in maize due to the knockdown of miR390, suggesting a broad impact on developmental and stress response pathways.

## 3. Discussion

Auxin plays a pivotal role in controlling plant morphology and environmental responses through its interaction with AUXIN RESPONSE FACTORS (ARFs) [41,42]. These factors are crucial in translating the auxin signal into developmental changes. Three miRNA families, miR160, miR167 and miR390, have been identified to confer plant growth via targeting or regulating different *ARFs*. In *Arabidopsis*, miR160 targets three *ARFs* genes, *ARF10*, *ARF16* and *ARF17* [43]; miR167 regulates *ARF6* and *ARF8* by mediating gene silencing [44]; miR390 negatively regulates the expression of *ARF2*, *ARF3*, and *ARF4* by triggering *TAS3*-tasiRNAs [45,46,47]. Additionally, miR165/166 and miR172 were also found to be involved in other phytohormone signaling, interacting with auxin signaling. miR165/166 was proved to play critical roles in ABA homeostasis [48] and auxin signaling [49]. miR172 was found to be involved in ethylene signaling [50]. A recent study revealed that the crosstalk of ethylene and auxin fine-tuned root gravitropic response [51]. In *Arabidopsis*, miR160 and miR167 regulate adventitious rooting by targeting *ARF17*, *ARF6,* and *ARF8* [52]; miR390 confers lateral root development through the *TAS3*-tasiRNA-*ARF* pathway [47]; the balance of miR165/166 and its targets play important roles in root development [53]; miR172 regulates plant de novo root regeneration through the ethylene signaling pathway [51]. Moreover, miRNAs such as miR444 [54], miR171 [55], miR393, and miR390 [56] have been identified to regulate root development in rice. In maize, several miRNAs were also identified in controlling maize root development, including miR172 [57], miR166 [25,26], and miR390 [26]. miR172 regulates maize brace root by suppressing the expression of its target gene *ZmRAP2.7* [57]. In this study, we found that the expression of miR167, miR172, miR166, and miR390 display significant alterations between the emergence and growth stages of maize brace root development. Together with these previous studies, our results indicate that the four miRNAs possibly play essential roles in regulating maize brace root growth.

The miR390-*TAS3*-tasiRNA-*ARF* regulatory module was found to play a pivotal role in regulating root development [26]. The underlying mechanism of miR390 in control of brace root development remains unclear. In the present study, the miR390 inactivation mutant, *STTM390*, displayed similar phenotypic alterations to the loss-of-function mutants of *ZmARF11* and *ZmARF26* (two downstream regulated genes of miR390) [32], which indicated that miR390 and the two *ARFs* are involved in regulating brace root growth. Such phenotypic alterations are probably controlled by the changes in gene expression. The knockdown of miR390 mediates the two *ARFs* and *TAS3g* up-regulation; mediates *TAS3c*, miR166, and its target genes down-regulation. These results suggested that the miR390-*TAS3*-Zm*ARF11/26* regulatory module confers brace root growth through a complicated molecular mechanism. Transcriptome analysis revealed miR390 to be associated with the extracellular region, cell wall macromolecule metabolic process, cell wall macromolecule biosynthetic process, cell wall polysaccharide biosynthetic process, cell periphery, cell death, response to ethylene, and plant hormone signal transduction. These pathways are probably at the downstream of miR390, mediating the phenotypic alterations in maize brace root.

In modern breeding of maize, the planting density has increased from 30,000 plants per hectare in the 1930s to >80,000 plants per hectare in the 2010s [2,58,59]. High-density planting increases the competition between plants for nutrients and water, leading to remodeling of the root system architecture for improving water and nitrogen use efficiency [60,61,62]. The modern hybrid and inbred lines have the steep root system [15,61,63]. The brace root number, whorls growing into the soil, and spread width are tightly associated with maize root lodging resistance [9,64,65]. However, steeper root angle is favorable for accessing subsoil water, enhancing drought tolerance, and improving nitrogen capture, while shallow root angle improves the capture of phosphorus from topsoil [66,67]. Thus, an ideal root system architecture must reach a balance between lodging resistance and nutrition uptake. In this study, four known miRNAs were identified to be involved in the regulation of maize brace root growth, including miR390, miR166, miR167, and miR390. The obtained *STTM390*, *arf11*, and *arf26* mutants all showed increased brace root number and brace root length, which are beneficial for enhancing root lodging resistance and nutrition uptake. These discoveries provide valuable genetic targets for enhancing maize traits, such as lodging resistance and optimized root architecture, which are vital for improved water and nutrient acquisition.

## 4. Materials and Methods

### 4.1. Vector Construction and Maize Transformation

In alignment with our previous study [36,68], the STTM390 binary expression vector was constructed following the established methods for STTM vector assembly [69]. Initially, STTM390 was cloned into the intermediate vector pOT2-poly-cis, yielding the recombinant vector pOT2-STTM390. Subsequently, the binary expression vector pTF101.1, with an additional PacI cleavage site, was employed for maize STTM transformation. The STTM390 insert from pOT2-STTM390 was subsequently cloned into the modified binary expression vector pTF101.1/PacI, resulting in the construct pTF101.1-STTM390. The final construct was confirmed through Spectinomycin resistance and DNA sequencing using the STTM common real primer (Appendix A). The constructed pTF101.1-STTM390 binary expression vector was transformed in maize inbred line C01 (Life Science and Technology Center of China Seed Group Co., Wuhan, China).

### 4.2. Transgenic Plant Screening, Genotyping, and Phenotyping

The transgenic maize plants harboring the STTM390 construct were initially screened by basta resistance and subsequently genotyped using the STTM common real primer (Appendix A). Quantitative RT-PCR (qRT-PCR) was utilized to determine the expression levels of miR390, its target *TAS3* precursors, and downstream *ARFs* (all the primers are listed in Appendix A). To ensure genetic consistency, *STTM390* plants were self-pollinated over three generations to obtain homozygous lines. Phenotypic evaluations were performed in three distinct environments. During the flowering stage, brace root samples from *STTM390* and control C01 plants were collected at the growth stage for qRT-PCR analysis; each sample consisted of three biological replicates, with each replicate comprising three plants from the same transformation event. Samples were immediately frozen in liquid nitrogen and stored at −80 °C for subsequent transcriptomic analysis.

Additionally, EMS mutants for *ZmARF11* and *ZmARF26*, specifically *arf11* (*EMS4-10a34d*) and *arf26* (*EMS4-1c4e8c*), were acquired from maizeEMSDB (http://maizeems.qlnu.edu.cn/) [70]. Positive EMS mutants were screened by a PCR-based method (all the used primers are listed in Appendix A). The screened positive EMS mutant plants were self-pollinated twice to reproduce and obtain the homozygous mutants. Phenotypic alterations in these mutants were subsequently assessed in field conditions.

### 4.3. Small RNA Deep Sequencing, Transcriptome Sequencing Analysis, and qRT-PCR

Field cultivation of the inbred line B73 was conducted up to the flowering stage. Brace root samples were collected from the emergence and growth stages, with each set comprising three replicates, each including three randomly selected plants. These samples underwent small RNA deep sequencing and transcriptome sequencing analysis. Total RNA from the brace root samples was extracted using Trizol reagent (Invitrogen, Carlsbad, CA, USA) and assessed by 1% agarose gel electrophoresis to ensure quality. RNA samples were then submitted to Berry Genomics (Beijing, China) for high throughput sequencing using the Illumina NovaSeq 6000 platform.

For small RNA library construction, RNA fragments ranging from 18 to 30 nt were isolated using 15% polyacrylamide gel electrophoresis. These fragments were subsequently ligated with 5′ and 3′ adaptors and used as templates for cDNA synthesis. PCR amplification was carried out for about 18 cycles to generate sufficient products for sequencing. Six small RNA libraries (across two developmental stages and three biological replicates) were prepared using the TruSeq Small RNA Sample Prep Kit (Illumina Technologies, San Diego, CA, USA). The raw reads were processed to trim adaptors and remove low-quality reads and poly(N) sequences. The quality of the cleaned reads was assessed by evaluating the Q20, Q30, and GC content. All clean reads, ranging from 18 to 40 nt, were mapped on the maize reference genome (B73 RefGen_V4.42) using HISAT2 V2.1.0 (default parameters setting) [71]. miRNA expression levels were normalized using the parameter transcripts per million (TPM). The differentially expressed miRNA was defined as |log2 fold change|> 1 and used a *p*-value < 0.05.

Transcriptomic sequencing involved preparing cDNA libraries from brace root samples of B73 at both developmental stages, as well as from C01 and *STTM390* during the growth stage, using the TruSeq RNA Sample Prep Kit (Illumina, San Diego, CA, USA). These libraries were sequenced on the Illumina NovaSeq 6000 platform. Transcriptomic sequencing data commenced with the trimming of adapters and low-quality bases using Trimmomatic V0.36 (FSU, Tallahassee, FL, USA) [72]. The clean reads were aligned to the maize reference genome (B73 RefGen_V4.42) using the HISAT2 V2.1.0 (default parameters setting) [71]. Transcript abundance was further calculated as fragments per kilobase of exon per million fragments mapped (FPKM). DEGs were identified using DESeq2 software V1.22.2 [73], with the threshold settings of |log2 fold change| ≥ 1 and *p*-values < 0.05. Gene ontology (GO) and Kyoto Encyclopedia of Genes and Genomes (KEGG) enrichment analyses were conducted using the maize profile database (org.Zeamays.eg.sqlite) in clusterProfiler software V3.10.1 [74] and Annotation Hub (V2.14.5) R package. The enrichment analyses used the ensemble database to convert the gene number (maizegbdId) to the corresponding Entrez ID.

To validate the sequencing results and measure the expression levels of miR390, miR166, their target genes, and downstream regulated genes, qRT-PCR analysis was conducted. Expression levels of miR390 and miR166 were quantified using the Mir-X™ miRNA qRT-PCR SYBR^®^ Kit (Takara, Dalian, China), with U6 small nuclear RNA serving as the internal control. For the protein-encoding genes, total RNA of the tested samples was reverse transcribed using the PrimeScript™ RT reagent kit with gDNA Eraser (Perfect Real Time) (Takara, Dalian, China). The resulting cDNA was used for qRT-PCR with the CFX96 Touch™ Real-Time PCR Detection System (Bio-Rad, Hercules, CA, USA) and the SYBR^®^ Premix EX Taq™ II (Tli RNaseH Plus) Kit (Takara, Dalian, China). Actin was used as an internal control, and the relative gene expression levels were calculated using the 2−ΔΔCt method [75]. All assays included at least three biological replicates. The qRT–PCR primers are listed in Appendix A.

### 4.4. Histological Analysis and Plant Hormone Content Measurement

For histological analysis, brace root samples from C01 and *STTM390* maize plants were collected during the flowering stage. The elongation zones of these samples, ap-proximately 1 cm in length, were excised and immediately submerged in a pre-chilled 70% FAA (formalin-acetic acid-alcohol) solution. These samples were then vacuum-sealed to remove air and fixed overnight at 4 °C in FAA. Following fixation, the samples underwent a graded ethanol dehydration series, were cleared, and then were embedded in paraffin wax using standard histological techniques (Sigma-Aldrich, St. Louis, MA, USA). The embedded tissues were sectioned into 4 μm slices using a Leica RM 2265 programmable rotary microtome (Leica Microsystems, Wetzlar, Hessian, Germany) and stained with 0.05% toluidine blue. The stained sections were imaged with an Olympus IX73 microscope (Olympus, Tokyo, Japan).

Additionally, the dried brace root samples from C01 and *STTM390* plants, previously prepared for qRT-PCR, were analyzed to quantify plant hormone levels. The indoleacetic acid (IAA) and abscisic acid (ABA) contents were determined using high-performance liquid chromatography (HPLC). These measurements were carried out for three biological replicates and four technical replicates, with hormone concentrations calculated based on the dry weight of the samples (Suzhou Comin Biotechnology Co. Ltd., Suzhou, China).

### 4.5. Statistical Analyses

All the collected data from phenotypic analysis and qRT-PCR analysis data were subjected to one-way variance analysis (ANOVA) and Student’s *t*-test using software SPSS 22.0 (IBM, Armonk, NY, USA). *p* < 0.05 indicates the statistical differences reach the significantly different level; *p* < 0.01 and *p* < 0.001 indicate very significantly different levels.

## Figures and Tables

**Figure 1 ijms-25-06791-f001:**
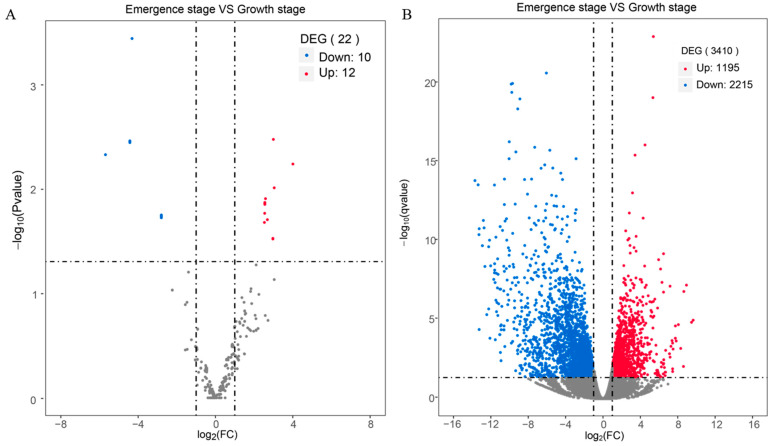
Small RNA-seq and transcriptome analysis of brace root during emergence and growth stage in B73. (**A**) Expression classification of differentially expressed miRNAs. Red dots indicate up-regulated miRNAs, and blue dots are down-regulated miRNAs (significance level: *p*-value < 0.05, |log2 fold change| > 1). (**B**) Gene expression classification of DEGs. Red dots indicate up-regulated genes, and blue dots are down-regulated genes (significance level: *p*-value < 0.05, |log2 fold change| > 1).

**Figure 2 ijms-25-06791-f002:**
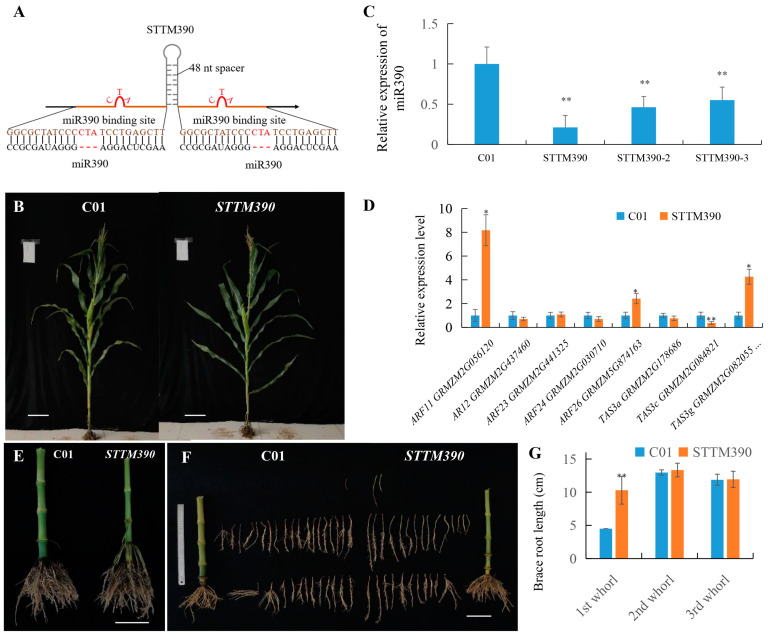
miR390 inactivation by STTM technology and phenotypic alterations in maize. (**A**) Diagram of STTM390 structure showing a 48 nt length spacer with two uncleavable miR390 binding sites, designed with three-base insertion (CTA) to resist cleavage by miR390. (**B**) Comparative plant phenotypes of *STTM390* and C01 plants, displaying noticeable differences in overall plant morphology. Bar = 20 cm. (**C**) qRT-PCR analysis of miR390 expression levels in *STTM390* and C01 plants. (**D**) Expression levels of miR390 target genes and downstream regulated genes in *STTM390* and C01 plants. (**E**–**G**) Phenotypic analysis of root systems in *STTM390* and C01 plants. Bar = 10 cm. The 1st whorl indicates the uppermost brace root whorl, the 2nd whorl is the second brace root whorl from the uppermost to the bottom, and the 3rd whorl is the third brace root whorl. U6 small nuclear RNA and *Actin* were used as the internal control of miR390 and target/downstream regulated genes in qRT-PCR analysis, respectively. *, ** represents that the corresponding miR390 and gene expression levels of *STTM390* plants are significantly and very significantly different from the wild type, *respectively*. Bars show standard error of tested values.

**Figure 3 ijms-25-06791-f003:**
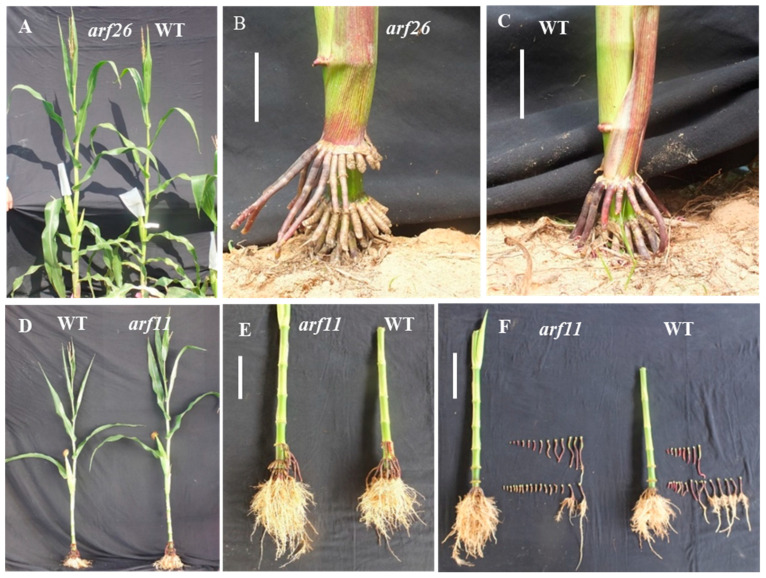
Comparative phenotypic analysis of *arf11* and *arf26* mutants in maize. (**A**) Phenotypic comparison between *arf26* mutant and WT. (**B**,**C**) Phenotypic difference of brace root between *arf26* mutant and WT. Bar = 10 cm. (**D**) Phenotypic comparison of plant traits between *arf11* mutant and WT. (**E**,**F**) Root system comparison between *arf11* mutant and WT maize. Bar = 10 cm.

**Figure 4 ijms-25-06791-f004:**
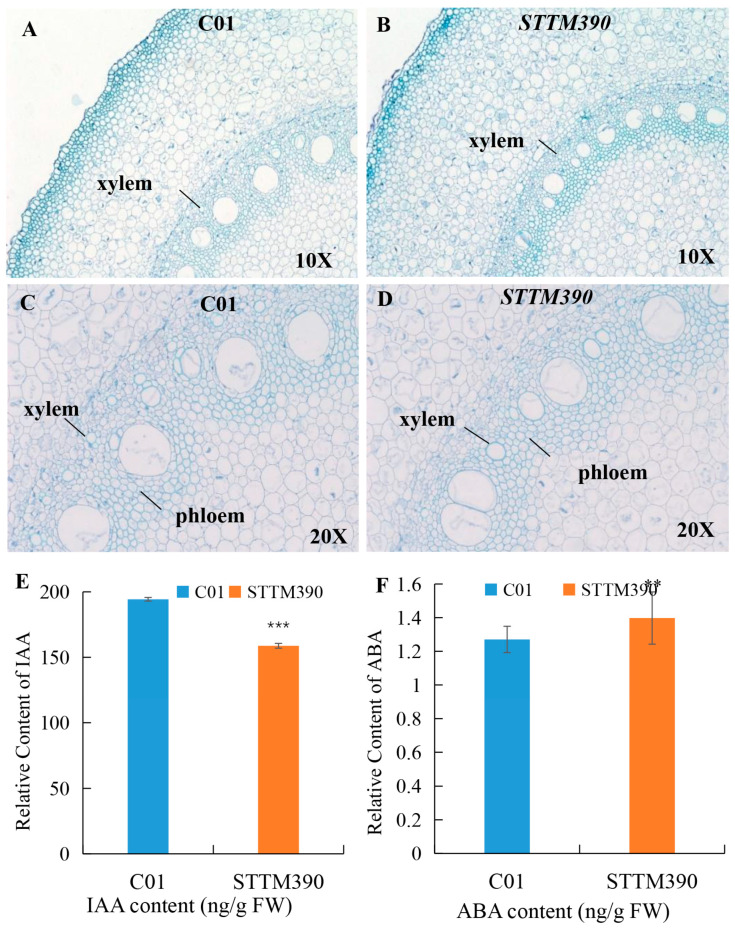
Histological analysis and IAA and ABA content measurement in *STTM390* and C01 brace roots. (**A**–**D**) Cross section views of plant brace root in *STTM390* and C01 (10× and 20×). (**E**,**F**) Quantitative comparison of Indole-3-acetic acid (IAA) and Abscisic acid (ABA) contents between *STTM390* and C01. **, *** represent a significant difference at *p* < 0.01 and *p* < 0.001 levels, respectively. Bars show standard error of tested values.

**Figure 5 ijms-25-06791-f005:**
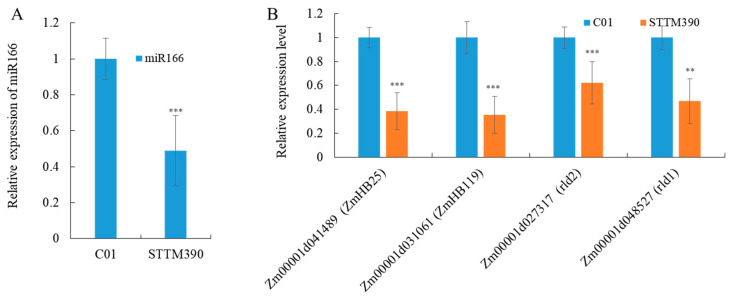
Relative expression level of miR166 and its target genes in *STTM390* brace roots. (**A**) The expression of miR166 in *STTM390* and C01 brace root. (**B**) The expression levels of four miR166 target genes in *STTM390* and C01 brace root. U6 and *Actin* were used as the internal control for miR166 and its target genes, respectively. ** and *** represents that the corresponding gene expression levels in *STTM390* plants are significantly different from the wild type at *p* < 0.01 and *p* < 0.001 levels, respectively. Bars show standard error of the tested values.

**Figure 6 ijms-25-06791-f006:**
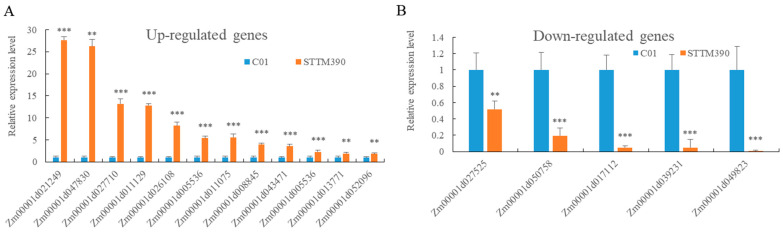
The expression analysis of DEGs between *STTM390* and C01 by RT-qPCR. (**A**) The expression of down-regulated genes. (**B**) The expression of up-regulated genes. *Actin* was used as an internal control. ** and *** represent that the corresponding gene expression levels in *STTM390* plants are significantly different from the wild type at *p* < 0.01 and *p* < 0.001 levels, respectively. Bars show standard error of the tested values.

**Figure 7 ijms-25-06791-f007:**
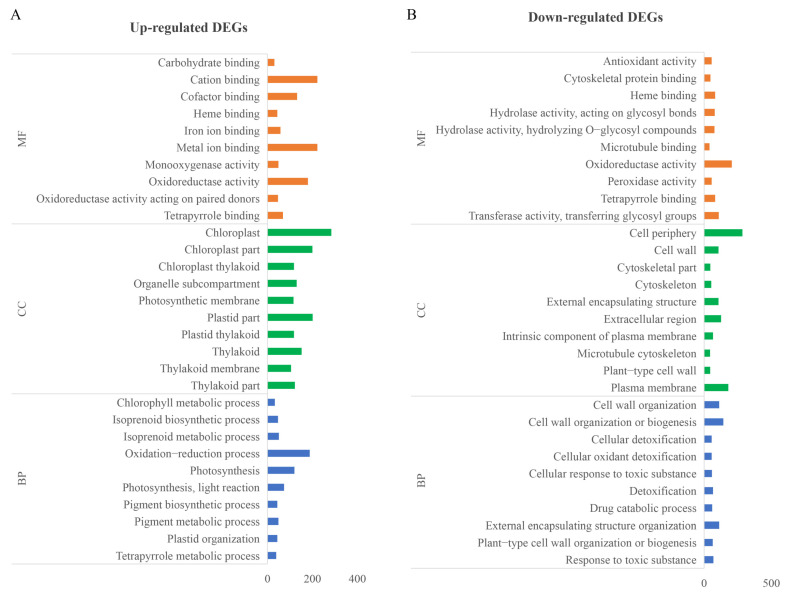
Gene ontology analysis of the screened DEGs between *STTM390* and C01 plants. (**A**) This panel displays the gene ontology (GO) categories for up-regulated DEGs. (**B**) This panel shows the GO categories for down-regulated DEGs. BP, biological process; CC, cell component; MF, molecular function.

**Figure 8 ijms-25-06791-f008:**
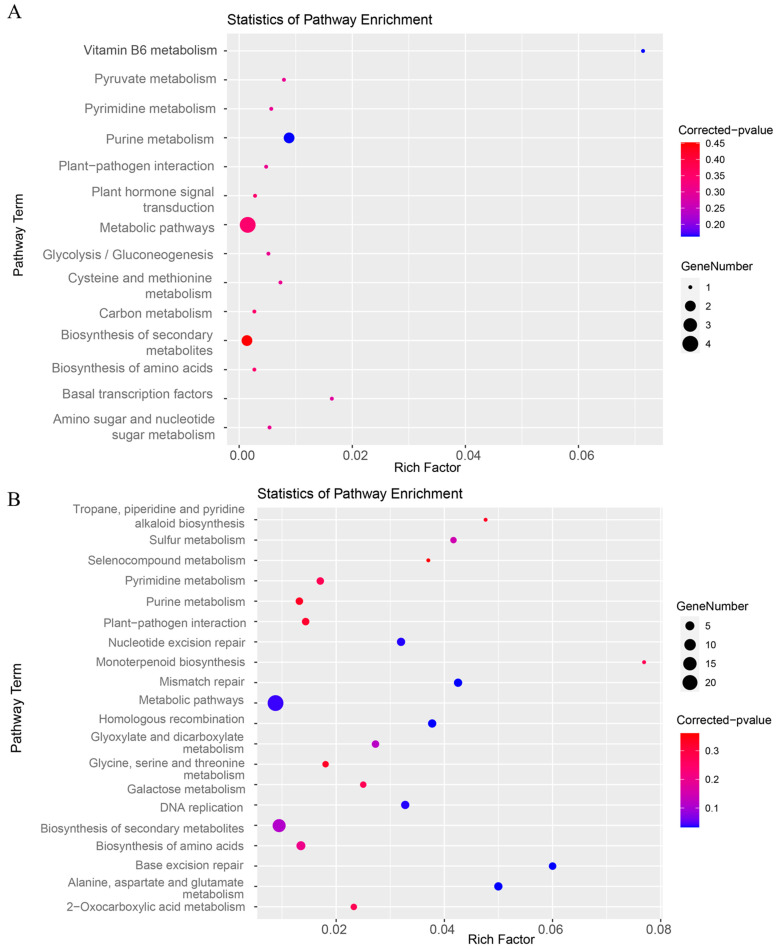
Kyoto encyclopedia of genes and genomes (KEGG) enrichment analysis of DEGs between *STTM390* and C01 plants. (**A**) Enrichment of down-regulated genes: This panel presents the KEGG pathway analysis for genes that are down-regulated in *STTM390* compared to the control. (**B**) Enrichment of up-regulated genes: This panel illustrates the KEGG pathway enrichment for up-regulated genes in *STTM390* plants.

**Table 1 ijms-25-06791-t001:** The screened brace root growth-related miRNAs by sRNA-seq analysis in B73.

miRNA	log_2_FoldChange	*p* Value
zma-miR167b-3p	4.01	5.80 × 10^−3^
zma-miR172d-5p	3.00	3.37 × 10^−3^
novel_chr7_31869	3.04	9.82 × 10^−3^
zma-miR390b-5p	2.97	2.99 × 10^−2^
zma-miR390a-5p	2.97	3.03 × 10^−2^
zma-miR166m-3p	2.68	1.98 × 10^−2^
zma-miR172b-5p	2.59	1.25 × 10^−2^
novel_chr4_18243	2.55	1.35 × 10^−2^
novel_chr5_20983	2.55	1.38 × 10^−2^
novel_chr1_2735	2.55	1.42 × 10^−2^
novel_chr7_30167	2.55	1.72 × 10^−2^
zma-miR166l-3p	2.53	2.11 × 10^−2^
novel_chr1_3716	−5.69	4.73 × 10^−3^
novel_chr7_30568	−4.31	3.64 × 10^−4^
novel_chrPt_42759	−4.43	3.49 × 10^−3^
novel_chr7_31057	−4.43	3.52 × 10^−3^
novel_chr6_27533	−4.43	3.53 × 10^−3^
novel_chr2_8182	−4.43	3.62 × 10^−3^
novel_chr9_37021	−2.80	1.79 × 10^−2^
novel_B73V4_ctg134_43715	−2.80	1.82 × 10^−2^
novel_chrPt_42651	−2.80	1.86 × 10^−2^
novel_chr5_25042	−2.80	1.90 × 10^−2^

**Table 2 ijms-25-06791-t002:** The screened key DEGs between brace root emergence and growth stages by transcriptomic analysis in B73.

ID	Emergence Stage—FPKM	Growth Stage— FPKM	log_2_FC	Description
*Zm00001d025533*	134.50	2.05	6.48	nad h-dependent oxidoreductase
*Zm00001d001960*	2000.73	22.98	3.65	fht1—flavanone 3-hydroxylase1
*Zm00001d014121*	658.50	23.51	5.85	cyp29—cytochrome P450 CYP71Z19
*Zm00001d042683*	291.51	1.82	5.55	protein nrt1 ptr family-like
*Zm00001d052683*	11,514.79	337.37	5.39	o-methyltransferase zrp4
*Zm00001d041842*	2605.69	72.83	5.35	aaap18—amino acid/auxin permease18
*Zm00001d014914*	1441.25	27.78	5.21	a2—anthocyaninless2
*Zm00001d007403*	743.55	18.41	5.14	whp1—white pollen1
*Zm00001d002755*	32.21	3.75	5.14	uncharacterized protein loc100280109 isoform x1
*Zm00001d044122*	4487.86	68.02	5.11	a1—anthocyaninless1
*Zm00001d015224*	79.53	4.80	5.01	aomt8—anthranilate O-methyltransferase8
*Zm00001d021991*	434.20	10.28	4.97	udp-galactose transporter 2-like
*Zm00001d033719*	194.64	6.14	4.83	col2—C2C2-CO-like-transcription factor 2
*Zm00001d006913*	101.60	2.06	4.67	trpp3-trehalose-6-phosphate phosphatase3
*Zm00001d035390*	189.61	11.25	4.65	scmv1—resistance to sugarcane mosaic virus1
*Zm00001d010630*	410.13	12.13	4.63	skus4—skewed root growth similar4
*Zm00001d023707*	159.01	6.14	4.58	trm1—thioredoxin M1
*Zm00001d039644*	172.72	10.29	4.50	cytokinin-o-glucosyltransferase 3
*Zm00001d009057*	794.02	39.30	4.50	pgl49—polygalacturonase49
*Zm00001d040468*	617.00	33.55	4.46	bz1—bronze1
*Zm00001d018260*	7.27	173.59	−4.56	growth-regulating factor
*Zm00001d020418*	0.68	53.32	−6.05	cytokinin hydroxylase-like
*Zm00001d052112*	7.22	552.8	−6.24	growth-regulating factor
*Zm00001d031594*	0.00	19.26	−7.28	probable auxin efflux carrier component 4
*Zm00001d050201*	1.90	501.36	−7.91	xyloglucan endotransglucosylase hydrolase protein 24-like
*Zm00001d049336*	0.16	5087.32	−8.09	probable xyloglucan glycosyltransferase 5
*Zm00001d043244*	0.00	40.28	−8.34	probable indole-3-acetic acid-amido synthetase
*Zm00001d040379*	0.02	268.88	−8.38	expansin-a11-like isoform x1
*Zm00001d025873*	1.82	812.82	−9.12	low-molecular-weight cysteine-rich protein lcr69 precursor
*Zm00001d045135*	0.01	291.93	−9.32	gibberellin 3-beta-dioxygenase 1
*Zm00001d043523*	3.70	438.73	−9.33	profilin a
*Zm00001d047113*	2.04	612.44	−9.40	Cortical cell-delineating protein
*Zm00001d054055*	0.01	20.48	−9.80	protein walls are thin 1-like
*Zm00001d039636*	0.02	781.09	−9.83	uncharacterized protein loc100821692
*Zm00001d024178*	0.02	1061.21	−10.05	probable xyloglucan glycosyltransferase 3
*Zm00001d044031*	0.00	139.56	−10.13	protein trichome birefringence-like 19
*Zm00001d045420*	1.24	4067.54	−10.54	gdp-mannose dehydratase 1-like
*Zm00001d011156*	0.00	188.01	−10.56	polygalacturonase *AT1G48100*
*Zm00001d026163*	7.36	3454.64	−11.62	cortical cell-delineating protein precursor
*Zm00001d015715*	0.02	11,179.29	−12.44	uncharacterized protein loc100272483 precursor

**Table 3 ijms-25-06791-t003:** The screened key DEGs between *STTM390* and C01 in the transcriptomic analysis in brace roots.

Gene	C01—FPKM	*STTM390*—FPKM	*log2FC*	Description
*Zm00001d024543*	8.33 × 10^−17^	5140.85	14.76	nac domain transcription factor superfamily protein
*Zm00001d043471*	2.18	344.35	9.07	transducin beta-like protein 3
*Zm00001d011129*	8.33 × 10^−17^	42.54	8.40	tubulin alpha-3 chain
*Zm00001d013771*	8.33 × 10^−17^	40.45	7.64	repressor of rna polymerase iii transcription maf1 homolog
*Zm00001d052096*	0.98	94.19	6.54	4-coumarate—ligase-like 1
*Zm00001d047830*	0.01	210.44	6.54	cytochrome p450 88a1-like
*Zm00001d027710*	0.02	157.12	6.02	peroxidase 44-like
*Zm00001d008845*	42.26	883.14	4.60	pre-mrna-processing-splicing factor 8-like
*Zm00001d021249*	1.96	156.99	4.54	bifunctional udp-glucose 4-epimerase and udp-xylose 4-epimerase 1-like
*Zm00001d011847*	13.27	177.74	3.74	bhlh54—bHLH-transcription factor 54
*Zm00001d003052*	9.21	116.41	3.66	nactf36—NAC-transcription factor 36
*Zm00001d016873*	102.84	1168.18	3.51	bhlh152—bHLH-transcription factor 152
*Zm00001d023669*	204.55	2013.20	3.30	nactf67—NAC-transcription factor 67
*Zm00001d037691*	5.09	30.70	2.59	bif4—barren inflorescence4
*Zm00001d026540*	20.02	99.30	2.31	arftf29—ARF-transcription factor 29
*Zm00001d048004*	1.93	8.70	2.16	ereb161—AP2-EREBP-transcription factor 161
*Zm00001d045203*	2.75	9.32	1.77	iaa41—Aux/IAA-transcription factor 41
*Zm00001d051891*	24.89	79.00	1.67	lbd24—LBD-transcription factor 24
*Zm00001d028919*	6.89	21.54	1.64	ereb26—AP2-EREBP-transcription factor 26
*Zm00001d030310*	430.55	1015.38	1.24	aic1—auxin import carrier1
*Zm00001d002025*	1535.63	699.80	−1.13	ereb24—AP2-EREBP-transcription factor 24
*Zm00001d033976*	14,369.66	7182.09	−1.00	iaa4—Aux/IAA-transcription factor4
*Zm00001d018414*	869.66	313.81	−1.47	iaa24—Aux/IAA-transcription factor 24
*Zm00001d026480*	14.29	3.53	−2.03	iaa44—Aux/IAA-transcription factor 44
*Zm00001d017112*	841.04	89.58	−3.23	b3 domain-containing protein os02g0598200-like isoform x1
*Zm00001d027525*	1235.24	23.30	−4.85	basic endochitinase c precursor
*Zm00001d039231*	100.08	3.04	−5.12	probable glucuronosyltransferase os06g0687900
*Zm00001d049823*	253.32	4.05	−6.27	mta sah nucleosidase
*Zm00001d050758*	31.44	8.33 × 10^−17^	−8.11	ribosome production factor 1
*Zm00001d021457*	68.80	8.33 × 10^−17^	−9.07	auxin-responsive protein saur36-like

## Data Availability

The raw transcriptomic sequencing data are available in the NCBI Sequence Read Archive (SRA) database (Accession number: PRJNA1097667, https://www.ncbi.nlm.nih.gov/sra/PRJNA1097667 (accessed on 9 April 2024)).

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
