# Peer review of "Knockdown of microRNA390 Enhances Maize Brace Root Growth"

_ijms, 2024, doi:10.3390/ijms25126791_

Round 1

Reviewer 1 Report

Comments and Suggestions for Authors

In this study, authors conducted small RNA sequencing to identify key microRNAs in the control of brace root growth. After that, they selected a miRNA to generate transgenic line using STTM technology. The found that microRNAs miR167, miR166, miR172, and miR390 regulate brace root growth in B73 inbred line. Knockdown of miR390 leads to elongated brace roots and increased whorls, with reduced IAA content and altered microstructure. Downstream genes ARF11 and ARF26 show upregulation, mirroring STTM390 phenotypes. This manuscript highlights miR390's pivotal role in brace root development, offering targets for genetic improvement and lodging resistance in maize.

Overall, the contents and purposes of the study are excellent for publication. It seems that manuscript is nicely written. However, the figures have poor quality and not well prepared for publication.

Please delete “.” from the numbers in all figures. For example, A -> A

In Figure 1A, what is BACK means of DEG?

In Figure 1 legend, please explain about Padj and Log2FoldChange in detail.

The quality of the graphs in the Figure 2 should be improved by changing the font colors in black. The Figure 2G, please explain about the 1st whorl 2nd whorl 3rd whorl in detail.

Figure 4E and 4F should be reorganized. Sizes are not equal between to graphs. Change font colors in black.

Figures 5, 6, 7, and 8, please improve quality of graphs by changing the font colors in black.

Line 414 and 431 p-value should be adjusted p-value.

Author Response

Dear the Editor and Reviewers,

Thank you very much for your critical comments for our manuscript. We have revised the manuscript carefully according to your suggestions. The major revisions are as following:

Overall, the contents and purposes of the study are excellent for publication. It seems that manuscript is nicely written. However, the figures have poor quality and not well prepared for publication.

We have improved the figure quality using different methods. In this revision, the figure quality is much improved.

Please delete “.” from the numbers in all figures. For example, A -> A

We have revised this point in all eight figures.

In Figure 1A, what is BACK means of DEG?

BACK indicates the expression levels of those miRNAs showing insignificant difference between the two tested brace root developmental phases, which is improper. We have revised this point.

In Figure 1 legend, please explain about Padj and Log2FoldChange in detail.

We have checked this point in Materials and Methods section, this description is a mistake. We have revised this mistake in manuscript.

The quality of the graphs in the Figure 2 should be improved by changing the font colors in black. The Figure 2G, please explain about the 1st whorl 2nd whorl 3rd whorl in detail.

The font color in Figure 2 is in black. However, when we saved the figure as PNG format, the font color become unsharpness. In Figure 2G, the 1st whorl indicates the uppermost brace root whorl, 2nd whorl for the second brace root whorl from the uppermost to bottom, and 3rd whorl for the third brace root whorl. We have added a description in the manuscript.

Figure 4E and 4F should be reorganized. Sizes are not equal between to graphs. Change font colors in black.

We have reorganized Figure 4 and improved the figure quality.

Figures 5, 6, 7, and 8, please improve quality of graphs by changing the font colors in black.

The font colors in the four figure are all in black. However, when we saved the figure as PNG format, the font color become unsharpness.

Line 414 and 431 p-value should be adjusted p-value.

We have revised this typing error.

Reviewer 2 Report

Comments and Suggestions for Authors

Major concerns

1. The authors showed that STTM390 transgenic plants with knockdown level of miR390 and up-regulated expression levels of ZmARF11 and ZmARF26 exhibited similar brace root phenotype with arf11 and arf26 mutant plants and concluded that ZmARF11 and ZmARF26 are downstream genes of miR390. The genetic evidence shown here did not support the conclusion. Please explain.

2. This manuscript is very descriptive and lack of well explanation and novelty.

Change title to “Knockdown of microRNA390 enhances maize brace root growth.”

English editing is required.

Abstract

The abstract needs to be changed in detail to be clearer and more informative.

Line 19: Identified what?

Introduction

Line 68: “brace” root growth not bract root growth

Line 80: remain largely unexplored. (in line 73: the increasing studies….yet in line 80: are poorly understood which are contradictory.)

Line 82: what ARF stands for? Do not use acronyms when used the first time in the manuscript.

Line 89: are still unclear.

Results

Line 110 and 112: green dots or blue dots?

Line 111 and 112 up-regulated and down-regulated genes or miRNAs in Figure 1B?

Line 146: what is the unit of the values shown in these two stages in Table 2? FPKM?

Line 148: how were the target genes of miR167 identified? Any analysis? Previous study? Is reference 42 the correct citation?

Line 163: in Result section 2.1, half of the genes in version 4 ID were italicized and the other half were not italicized. Please use the correct nomenclature and stay with it.

Line 172: Figure 2B: the wild-type plants exhibited narrow leaf angle and what about the transgenic STTM390 plants?

Line 175: Which STTM390 line was used for the phenotypic measurement and gene expression test in Figure 2B-G? Also, the images shown in Figure 2 look blurry. Please provide high quality images.

Line 178: increased brace root whorls? There is no clear increase.

Line 218: reference(s)?

Discussion

Line 294-296: references?

Do not repeat the introduction and result sections in the discussion.

Comments on the Quality of English Language

Extensive editing of English language required

Author Response

Dear the Editor and Reviewers,

Thank you very much for your critical comments for our manuscript. We have revised the manuscript carefully according to your suggestions. The major revisions are as following:

Major concerns

  1. The authors showed that STTM390transgenic plants with knockdown level of miR390 and up-regulated expression levels of ZmARF11 and ZmARF26 exhibited similar brace root phenotype with arf11 and arf26 mutant plants and concluded that ZmARF11 and ZmARF26 are downstream genes of miR390. The genetic evidence shown here did not support the conclusion. Please explain.

Previous studies revealed that maize miR390 family has two members, miR390a and miR390b, which cleave TAS3 precursors and generate tasiRNAs (Gautam et al., 2021). Then, TAS3-tasiRNAs further to target several ARFs, including ZmARF11, ZmARF12, ZmARF23, ZmARF24, and ZmARF26 (López-Ruiz). In the present study, STTM390 transgenic plants with knockdown level of miR390 and up-regulated expression levels of ZmARF11 and ZmARF26 exhibited similar brace root phenotype with arf11 and arf26 mutant plants, which indicated that miR390 and its downstream regulated genes ZmARF11 and ZmARF26 are involved in brace root growth. However, the underlying molecular mechanism still need to be further explored. We have revised the conclusion in manuscript.

Gautam, V.; Singh, A.; Yadav, S.; Singh, S.; Kumar, P.; Sarkar Das, S.; Sarkar, A.K. Conserved LBL1-ta-siRNA and miR165/166-RLD1/2 modules regulate root development in maize. Development 2021, 148, doi:10.1242/dev.190033.

López-Ruiz, B.A.; Juárez-González, V.T.; Gómez-Felipe, A.; De Folter, S.; Dinkova, T.D. tasiR-ARFs Production and Target Regulation during In Vitro Maize Plant Regeneration. Plants (Basel) 2020, 9, doi:10.3390/plants9070849.

  1. This manuscript is very descriptive and lack of well explanation and novelty.

In the present study, miR390 and its two downstream regulated genes were identified to play important roles in brace root growth. However, STTM390 and arf11 and arf26 mutant plant displayed the similar phenotypes, suggesting the underlying molecular mechanisms are much complicated. In STTM390 plants, TAS3c exhibited a significant decreased expression, whereas TAS3g expression was upregulated, suggesting a complex regulation by miR390 on different TAS3 precursors. Additionally, the inactivation of miR390 meditates down-regulation of miR166 and its target genes. Such a complicated regulatory mechanism is far from addressed.

Change title to “Knockdown of microRNA390 enhances maize brace root growth.”

 We have revised the title to “Knockdown of microRNA390 enhances maize brace root growth.”

English editing is required.

The Co-author Dr. Guiliang Tang have revised the Englis writing in detail. He have stayed in USA for twenty years.  

Abstract

The abstract needs to be changed in detail to be clearer and more informative.

We have revised the abstract section.

Line 19: Identified what?

We have revised this point.

Introduction

Line 68: “brace” root growth not bract root growth

We have revised this typing error.

Line 80: remain largely unexplored. (in line 73: the increasing studies….yet in line 80: are poorly understood which are contradictory.)

We have revised the description in the manuscript.

Line 82: what ARF stands for? Do not use acronyms when used the first time in the manuscript.

ARF stands for Auxin Response Factor. We have revised this point.

Line 89: are still unclear.

We have revised this mistake.

Results

Line 110 and 112: green dots or blue dots?

It should be blue dots. We have revised this mistake.

Line 111 and 112 up-regulated and down-regulated genes or miRNAs in Figure 1B?

They should be up-regulated and down-regulated genes. We have revised this mistake.

Line 146: what is the unit of the values shown in these two stages in Table 2? FPKM?

The unit of the values is FPKM. We have added this information in the manuscript.

Line 148: how were the target genes of miR167 identified? Any analysis? Previous study? Is reference 42 the correct citation?

The reference 42 is an incorrect citation. We have revised this mistake in the manuscript. In the study by Liu et al., maize miR167 targets were predicted using two online tools: WMD3 (http://wmd3.weigelworld.org/cgi-bin/webapp.cgi) and TargetMiRna (http://www.softberry.com/berry.phtml) based on B73_Refgen_V4 (https://plants.ensembl.org/Zea_mays/Info/Index). ZmARFs 3, 9, 16, 18, 22, 30, and 34 were identified contain ZmamiR167 binding sites.

Line 163: in Result section 2.1, half of the genes in version 4 ID were italicized and the other half were not italicized. Please use the correct nomenclature and stay with it.

We have revised such typing errors in the manuscript.

Line 172: Figure 2B: the wild-type plants exhibited narrow leaf angle and what about the transgenic STTM390 plants?

The obtained STTM390 transgenic mutant plants exhibited notable phenotypic changes, including reduced plant height, narrower leaf angles above the uppermost ear, and broader leaf angles below the uppermost ear compared to wild-type plants, which typically show narrow leaf angles below the uppermost ear. We have revised the description in the manuscript.

Line 175: Which STTM390 line was used for the phenotypic measurement and gene expression test in Figure 2B-G? Also, the images shown in Figure 2 look blurry. Please provide high quality images.

In the present study, we screened seven STTM390 lines from twenty transformation events. Two lines displayed serious defects in tassel development, we have not reproduced enough seeds for experiment. The rest five lines showed similar phenotypes, and we obtained enough seeds for this study. For the phenotypic measurement and gene expression test, we planted five STTM390 lines. At final, three out of the five lines were selected for investigation and sampling.

We have improved the Figure 2.

Line 178: increased brace root whorls? There is no clear increase.

As growing in local environments, most maize inbred lines and hybrids usually have two or three brace root whorls. In the present study, most of STTM390 plants have three brace root whorls, two whorls for C01 plants. Such increase is relative notable.

Line 218: reference(s)?

We have cited the related references.

Discussion

Line 294-296: references?

We have cited the related references.

Do not repeat the introduction and result sections in the discussion.

We have revised the discussion section in detail.

Reviewer 3 Report

Comments and Suggestions for Authors

The submitted MS reflects 45% plagiarism ...Thus, summarily rejected with a chance to resubmit after rewriting the paper. Authors are requested to remove the refence section while iThenticating it....Also, the methods section needs a complete rewriting.  

Comments on the Quality of English Language

NA

Author Response

Dear Editor and the Reviewer,

Thank you very much for your critical comments for our manuscript. We have rewritten the introduction, results, materials and methods sections. And, we have revised discussion section in detail.

Round 2

Reviewer 1 Report

Comments and Suggestions for Authors

Authors diligently carried out revisions according to reviewers' feedback. Now, the manuscript is suitable for publication in its current form. Congratulations on excellent work!

Reviewer 2 Report

Comments and Suggestions for Authors

The authors have addressed my major concerns although the mechanism behind miR390-mediated brace root growth remain elusive.

This revised version of manuscript is much cleaner and improved.